# Forest and Land Fires Are Mainly Associated with Deforestation in Riau Province, Indonesia

**Hari A. Adrianto** [1,2,*], **Dominick V. Spracklen** [1], **Stephen R. Arnold** [1], **Imas S. Sitanggang** [2] **and Lailan Syaufina** [3]

[1]    School of Earth and Environment, University of Leeds, Leeds LS2 9JT, UK;
       D.V.Spracklen@leeds.ac.uk (D.V.S.); S.Arnold@leeds.ac.uk (S.R.A.)
[2]    Computer Science Department, IPB University, Bogor 16680, Indonesia; imas.sitanggang@apps.ipb.ac.id
[3]    Department of Silviculture, Faculty of Forestry, IPB University, Bogor 16680, Indonesia;
       lailans@apps.ipb.ac.id
*    Correspondence: eehaa@leeds.ac.uk

**Abstract:** Indonesia has experienced extensive land-cover change and frequent vegetation and land fires in the past few decades. We combined a new land-cover dataset with satellite data on the timing and location of fires to make the first detailed assessment of the association of fire with specific land-cover transitions in Riau, Sumatra. During 1990 to 2017, secondary peat swamp forest declined in area from 40,000 to 10,000 $km^2$ and plantations (including oil palm) increased from around 10,000 to 40,000 $km^2$. The dominant land use transitions were secondary peat swamp forest converting directly to plantation, or first to shrub and then to plantation. During 2001–2017, we find that the frequency of fire is greatest in regions that change land-cover, with the greatest frequency in regions that transition from secondary peat swamp forest to shrub or plantation (0.15 $km^{-2}$ $yr^{-1}$). Areas that did not change land cover exhibit lower fire frequency, with shrub (0.06 $km^{-2}$ $yr^{-1}$) exhibiting a frequency of fire >60 times the frequency of fire in primary forest. Our analysis demonstrates that in Riau, fire is closely connected to land-cover change, and that the majority of fire is associated with the transition of secondary forest to shrub and plantation. Reducing the frequency of fire in Riau will require enhanced protection of secondary forests and restoration of shrub to natural forest.

**Keywords:** forest and land fire; land-cover transition; Riau Indonesia

## 1. Introduction

Vegetation and peat fires in Indonesia are a major environmental hazard. Fires emit substantial amounts of $CO_2$ and contribute to climate change. In 2015, fires were estimated to have emitted around 700–800 Tg $CO_2$ [1,2]. Trace gas and particulate emissions from fire cause regional air pollution [3]. In September and October 2015, over 60 million people in Sumatra, Borneo, Malaysia and Singapore were exposed to poor air quality from fires [4], contributing to 10,000–100,000 premature deaths [4,5]. Indonesia contains large areas of peatland. When fires burn on peat, they can burn deep into organic soils resulting in substantial emissions [6]. During the 2015 fires in Indonesia, peat burning contributed 55% of $CO_2$ emissions and 70% of primary fine particulate matter emissions from fires [1].

In the wet tropics where annual mean rainfall is >1500 mm, fire is normally a rare occurrence [7]. In Indonesia, fires are more common in dry years associated with positive ENSO index (El Niño) [8], but in recent years fires also occur even in non-drought years [9]. The clearing of forests [10–12] and drainage of peatlands, largely to establish oil palm and acacia plantations [13], has made the landscape more susceptible to fire. Fire often occurs in forested regions that are experiencing land-cover change [14]. Fire frequency is typically higher in oil palm and wood fibre concessions compared

to protected areas [15,16]. Fire is used as part of the land-conversion process, to clear vegetation in preparation for agriculture and plantations [17]. In Riau, Indonesia, fires are six times more frequent in regions experiencing recent tree cover loss compared to regions with no loss [16].

Understanding the links between land-cover change and fire is necessary to inform land and fire management and fire suppression efforts. However, there is still poor understanding of the fraction of fire that is associated with specific land-cover changes. Satellite datasets provide some information on land-cover change (i.e., canopy cover loss), but there is rarely detailed information on the specific land-cover transitions that occur. Here we combine a new land-cover dataset with information on the location and timing of fires from satellite, to make the first assessment of the association between fire and specific land-cover transitions in Indonesia. We focus on Riau province, one of the most active areas of fire in Indonesia.

## 2. Materials and Methods

Our study area consists of the province of Riau, Sumatra, covering 89,691 km$^2$ and consisting of 43% peatland [16]. We used the land-cover map provided by the Indonesian Ministry of Environment and Forestry (http://webgis.menlhk.go.id:8080/pl/pl.htm, [18]). The map includes land-cover classifications for 1990, 1996, 2000, 2003, 2006 and 2009, then annually between 2011 and 2017. Before 2000, the land-cover classification was conducted as a part of National Forest Inventory (NFI) project which predominantly relied on analysis of Landsat imagery. During 2000–2009, digital Landsat images were combined with 1000 m SPOT Vegetation and 250 m MODIS images, but the classification still depended on visual image interpretation. Finally, since 2009 only Landsat images have been used as main data source and Landsat 8 OLI have been used since 2013. The land-cover dataset includes 31,785 polygons, with land-cover divided into 23 different land-cover classifications (Table A1) which we use to form nine grouped land-cover classes (Table 1). We also used data on the location of concession areas (wood fibre, logging, and oil palm plantation) and protected area extents in 2010 provided by the World Resources Institute (http://data.globalforestwatch.org/datasets). Concessions include oil palm, wood fibre, and logging concessions.

**Table 1.** Land-cover classes, showing how we grouped land-cover types from Ministry of Environment and Forestry.

| Grouped Land-Cover | Code | Original Land-Cover Types and Code |
|---|---|---|
| Primary dryland forest | PDF | Primary dryland forest (2001) |
| Primary peat swamp forest | PSF | Primary swamp forest (2005), Primary mangrove forest (2004) |
| Secondary dryland forest | SDF | Secondary dryland forest (2002) |
| Secondary peat swamp forest | SSF | Secondary swamp forest (20051), Secondary mangrove forest (20041) |
| Plantation | PLT | Plantation forest (2006), Estate crop (2010) |
| Shrub | SRB | Non-Forest Dry shrub (2007), Wet shrub/swampy shrub (20071), Savanna and Grasses (3000), Bareground/Bare soil (2014) |
| Water | WTR | Fish pond/aquaculture (20094), Open water (5001), Open swamp (50011) |
| Agriculture | AGR | Dry Agriculture (20091), Mixed dry agriculture (20092), Paddy Field (20093) |
| Urban | URB | Settlement areas (2012), Port and Harbor (20121), Transmigration Area (20122), Mining_Area (20141) |

Information on the distribution of fire is available from thermal anomaly (active fire) products and burned area observations. Small fires that are below the detection limit of burned areas products can contribute 60% of total burned area in Equatorial Asia [19]. Here we used data from active fire products which provide more accurate data on the distribution of small fires [19]. The occurrence of fires was obtained from MODIS (Moderate Resolution Imaging Spectro-radiometer) on Terra/Aqua Satellites. The instrument has a spatial resolution of 1 km$^2$ resolution in the nadir [20]. We used the MCD14ML

Global Monthly Fire Location Product Collection 6, with a minimum detection size of ≈50 m² fires under pristine conditions [21]. This product has 1.2% global daytime commission error [20] and is suitable in describing the spatial arrangement of fire over various vegetation types [22]. In addition, MCD14ML may act as a good predictor for small burned area [23]. In this research, hotspots during 2001–2017 were obtained from https://firms.modaps.eosdis.nasa.gov/download/. The Indonesian Ministry of Environment and Forestry classify hotspots pixel based on their confidence level as low confidence (<30%), medium confidence (30–79%) and high confidence (80–100%). [24]. We followed this procedure and only analysed high confidence hotspots. We defined fire frequency as the number of hotspots detected per unit area per year ($km^{-2}$ $year^{-1}$).

We used the land-cover dataset to identify regions that have experienced land-cover transitions and regions that have not changed land cover. For 2003–2017 when we had overlapping information on land-cover and active fires, we calculated the fire frequency for different land-covers and land-cover transitions.

## 3. Results

Between 1990 and 2017 there has been a steady decline in natural forest cover in Riau and an expansion of plantation (PLT), shrub (SRB) and agriculture (AGR) (Figure 1). Secondary peat swamp forest (SSF) declined from an area of around 40,000 km² in 1990 to around to 10,000 km² in 2017. The rate of loss of SSF was fairly constant between 1990 and 2012, with slower loss between 2012 and 2017. Secondary dryland forest (SDF) also decreased, from around 15,000 km² to less than 3000 km² in 2017. Primary forest was already quite diminished in 1990, with only 2133 km² of primary swamp forest (PSF) and 1648 km² of primary dryland forest (PDF) remaining. Primary forest decreased further between 1990 and 2017, with the area of PSF decreasing to 562 km² and the area of PDF reducing to 1502 km². The area of plantation increased steadily, from around 10,000 km² in 1990 to around 40,000 km² in 2017. The area of shrub increased from around 10,000 km² in 1990 to a maximum of around 23,000 km² in 2012, before declining to around 15,000 km² in 2017. Agriculture also expanded from around 10,000 km² in 1990 to 15,000 km² in 2017.

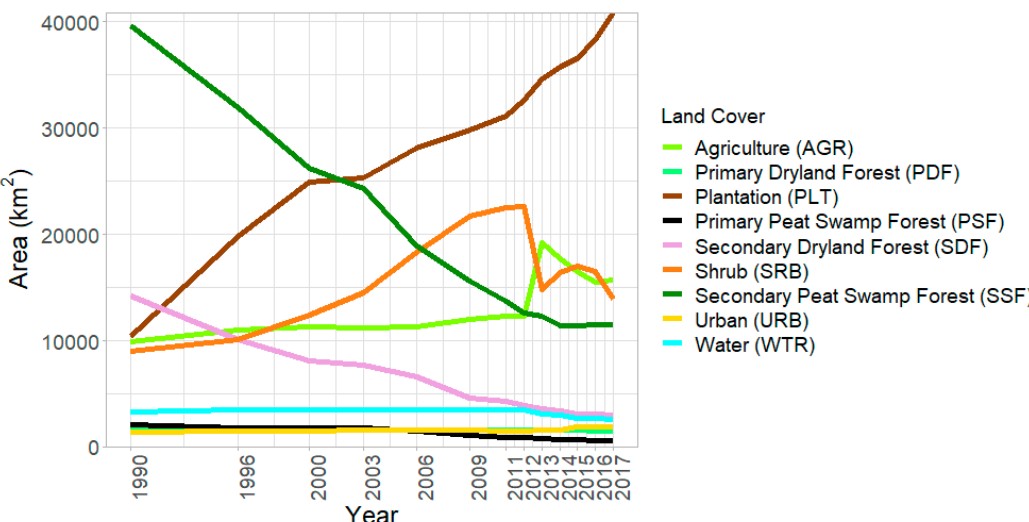

**Figure 1.** Land-cover change across Riau between 1990 and 2017. Land-cover classes are described in Table 1.

Figure 2 shows the major temporal (4–6 years gap) land-cover transitions that have occurred over this period. Secondary forest (dryland SDF and swamp SSF) has primarily been converted into plantation both directly (1996–2012) or via a transition to shrub then to plantation (2012–2017). Notably, there is less conversion of secondary forest since 2012.

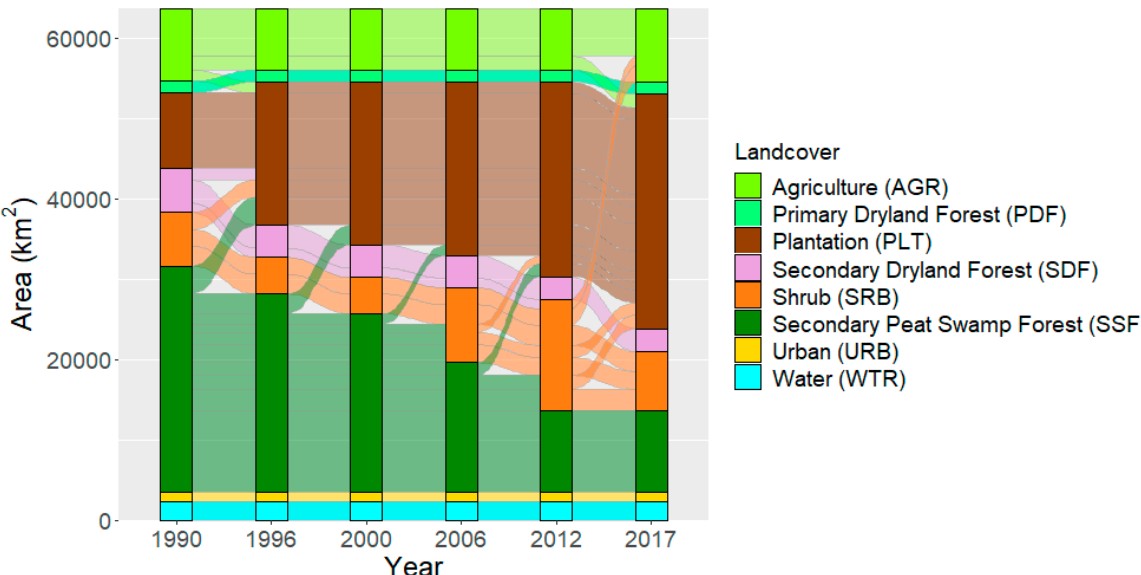

**Figure 2.** Major land-cover transitions in Riau.

Figure 3 shows the timings of the major land-cover transitions. The largest transitions were SSF to plantation (12,285 km$^2$) and shrub (14,611 km$^2$) and shrub to plantation (11,092 km$^2$) (Table 2). In 1990–1996, SSF declined from 43 to 35% (Figure 3b), converted to plantation (3781 km$^2$), shrub (2286 km$^2$) and agriculture (1700 km$^2$) (Table 2). Between 1996 and 2006, SSF declined from 35 to 21%, converted into shrub (8175 km$^2$) and plantation (4523 km$^2$) (Figure 3c). During 2006–2017, the largest conversions were shrub to agriculture (5000 km$^2$) and shrub to plantation (6000 km$^2$) (Figure 3d, Table 2).

**Table 2.** Summary of the area of the main land-cover transitions (km$^2$) in Riau.

| Initial Type | End Type | Transition | 1990–1996 | 1996–2000 | 2000–2006 | 2006–2017 | Sum |
|---|---|---|---|---|---|---|---|
| Secondary Peat Swamp Forest | Plantation | SSF→PLT | 3781 | 2698 | 1825 | 3981 | 12,285 |
| | Shrub | SSF→SRB | 2286 | 2462 | 5713 | 4150 | 14,611 |
| | Agriculture | SSF→AGR | 1700 | 485 | 76 | 487 | 2748 |
| Secondary Dryland Forest | Plantation | SDF→PLT | 2012 | 738 | 235 | 1259 | 4244 |
| | Shrub | SDF→SRB | 1422 | 981 | 1182 | 615 | 4200 |
| | Agriculture | SDF→AGR | 583 | 347 | 77 | 1751 | 2758 |
| Shrub | Plantation | SRB→PLT | 2417 | 1016 | 1912 | 5747 | 11,092 |
| | Agriculture | SRB→AGR | 376 | 205 | 54 | 4742 | 5377 |

Figure 4 shows the average fire frequency, both for regions that have not changed land-cover type and for areas that have changed land-cover. Results were analysed for the period 2003 to 2017, when we had overlapping data on land-cover and active fires. Regions that do not change land-cover type have lower fire frequency (<0.025 km$^{-2}$ yr$^{-1}$ except in shrub), compared to regions that experience a land-cover transition (up to 0.15 km$^{-2}$ yr$^{-1}$). The greatest fire frequency occurred in secondary peat swamp forest converted to shrub or plantation ($\approx$0.14 km$^{-2}$ yr$^{-1}$), shrub converted to plantation (0.1 km$^{-2}$ yr$^{-1}$) and secondary dry forests converted to plantation (0.09 km$^{-2}$ yr$^{-1}$) or agriculture (0.06 km$^{-2}$ yr$^{-1}$). Of the regions that changed land-cover type, agriculture to plantation had the lowest fire frequency (0.02 km$^{-2}$ yr$^{-1}$), likely because there was less need to use fire to clear vegetation from the land during this conversion.

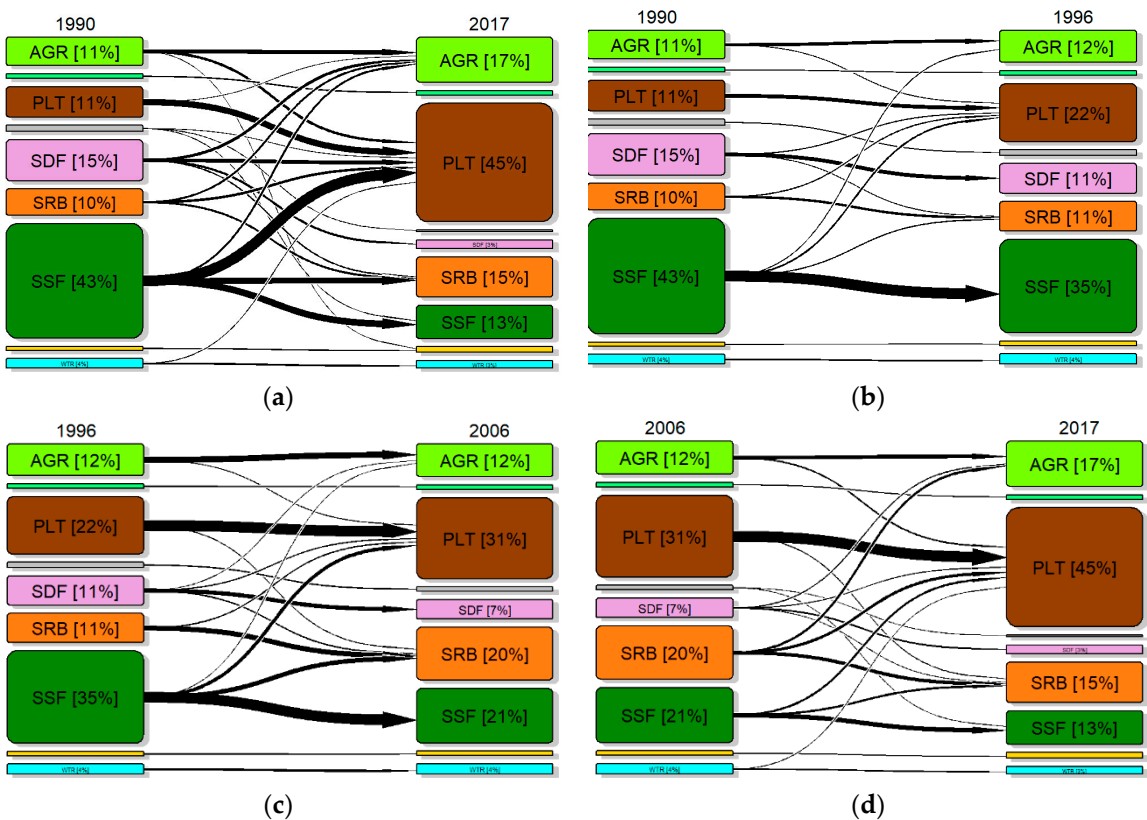

**Figure 3.** Land cover transitions occurring between (**a**) 1990 and 2017, (**b**) 1990 and 1996, (**c**) 1996 and 2006, (**d**) 2006 and 2017. Land cover codes are AGR: Agriculture, PDF: Primary Dryland Forest, PLT: Plantation, PSF: Primary Peat Swamp Forest, SDF: Secondary Dryland Forest, SRB: Shrub, SSF: Secondary Peat Swamp Forest.

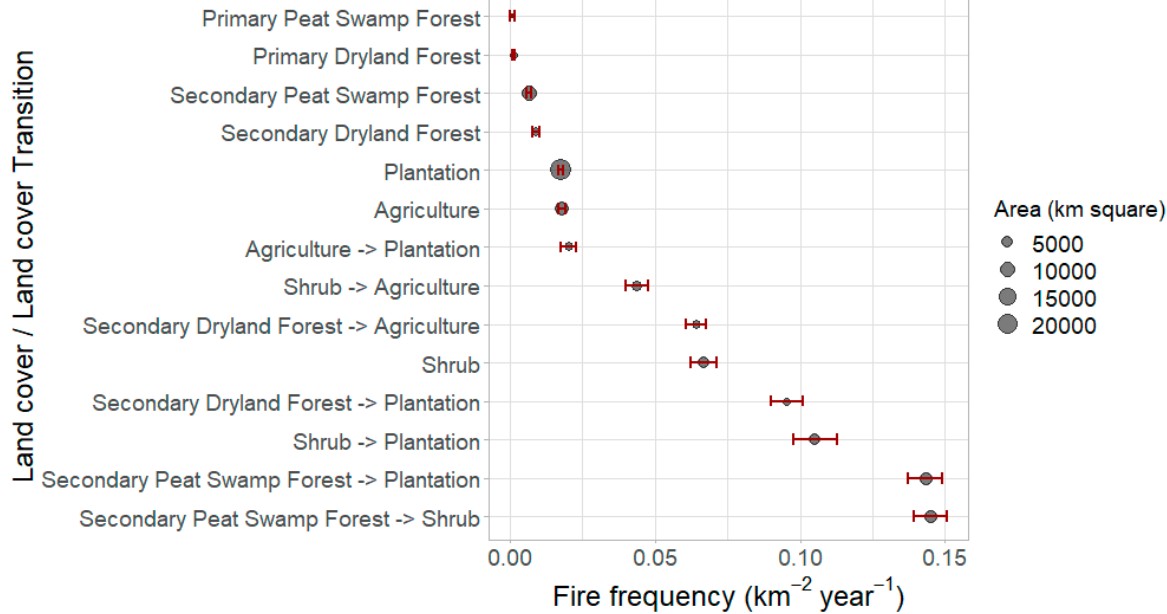

**Figure 4.** Average fire frequency for different land-covers and land-cover transitions between 2003 and 2017. Average fire frequency (point) and their 95% confidence interval (bar). Size of point shows the area of land-covers (detailed in Table 1) and land-cover transitions (detailed in Table 2).

Within regions that do not change land-cover, shrub (0.067 km$^{-2}$ yr$^{-1}$) has the greatest fire frequency, several times higher than agriculture (0.018 km$^{-2}$ yr$^{-1}$) or plantation (0.017 km$^{-2}$ yr$^{-1}$). Primary wet and primary dry forests experience a very low fire frequency (0.001 km$^{-2}$ yr$^{-1}$), a factor of 67 less than experienced in shrub regions and a factor of 17–18 less than in plantation or agriculture. Secondary dry and secondary peat swamp forest also experience low fire frequency (0.009 and 0.006 km$^{-2}$ yr$^{-1}$, respectively), a factor of 7 less than shrub and half the frequency experienced in agriculture or plantation.

Figure 5 shows how fire frequency has changed over time both for regions that did not experience a land-cover transition (Figure 5a) and regions that did (Figure 5b). Over 2002–2016, fire frequency has declined in agriculture and plantation land covers but has increased in secondary swamp and secondary dry forests. This may possibly indicate an increasing degradation of secondary forests over this period, increasing the potential for fire. Fire frequency in primary forests has remained very low over the whole period. Figure 5 emphasizes the risk of fire in shrub, since all land cover that suffers hotspot density larger than 0.1 km$^{-2}$ yr$^{-1}$ involved shrub. However, shrub areas which changed into plantation or agriculture had lower fire frequency after the land-cover transition. For land-cover transitions involving conversion of secondary forest to shrub, the greatest fire frequency typically coincides with the timing of the land-cover transition. After the land-cover has transitioned to shrub, fire frequency remains enhanced demonstrating a permanent transition to a more fire-prone state.

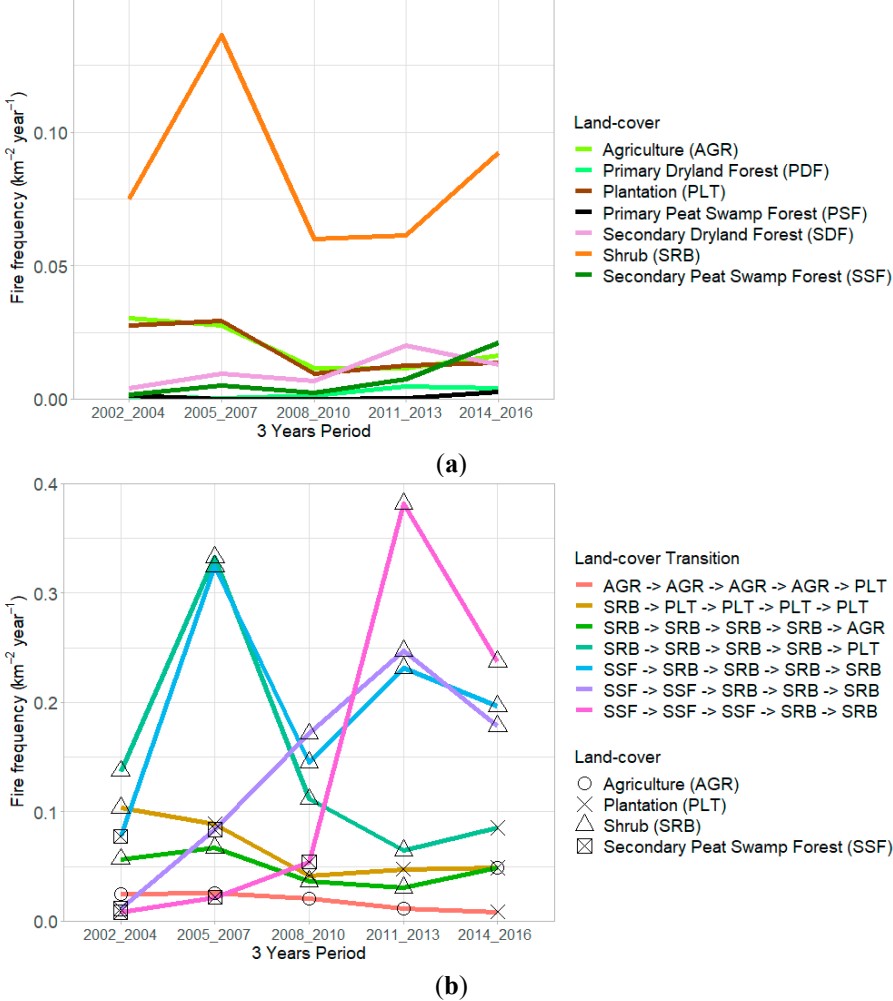

**Figure 5.** Frequency of fire according to (**a**) land-cover type and (**b**) land-cover transition, for the largest land-cover transitions (>1000 km$^2$). Land-cover taken from year in the centre of period. Hotspot density is calculated as the average of three years surrounding the land-cover transition.

Figure 6 shows the frequency of fire across land-cover types and transitions, separately for different land-use concessions. In shrub areas, the greatest fire frequency occurs in oil palm and wood fibre concessions. In secondary forests the greatest fire frequency also occurs in oil palm concessions. In shrub areas that were converted to plantation, the most frequent fires occur in oil palm concessions and areas of land outside of concessions or protected areas (Other).

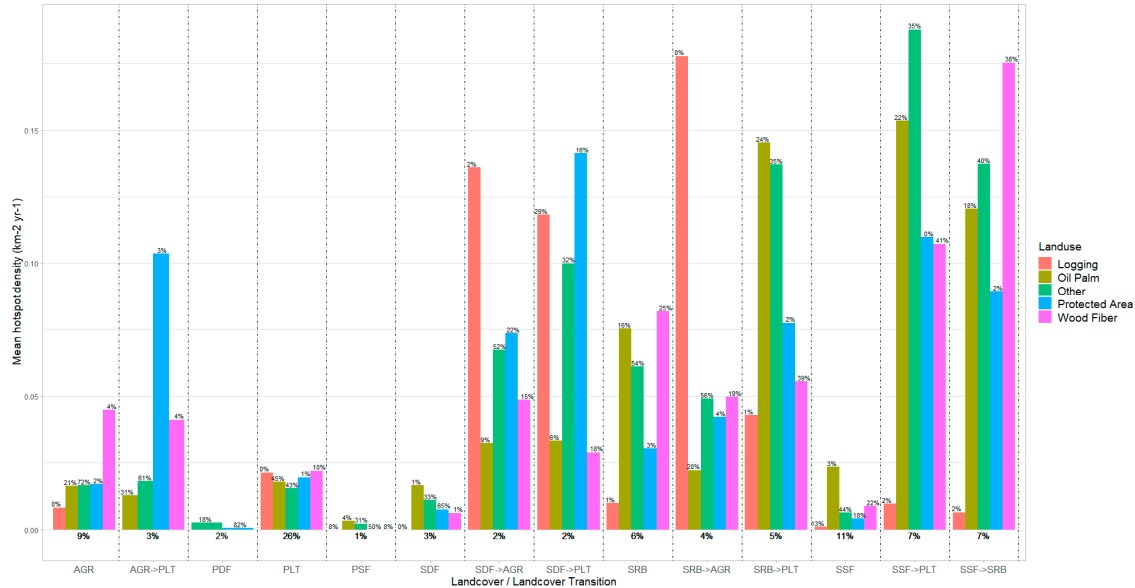

**Figure 6.** Average hot spot density (km$^{-2}$ yr$^{-1}$) for different land-cover and land-cover transitions between 2003 and 2017, separated for different land-use concessions.

## 4. Discussion

Our analysis shows that the greatest fire frequency in Riau Province, Sumatra, occurs in regions that have been converted from secondary peat swamp forest to shrub and plantation. Previous studies have shown fire in Indonesia often occurs in forested regions [14]. Our analysis demonstrates that this is linked to transition of forest to other land-covers. Highest fire frequency coincides with timing of the land-cover transition, confirming that fire is often used to carry out land use changes [25].

Shrub experienced the highest fire frequency, with lower fire frequency in plantation or agriculture areas. We find that shrub is often a transition land-cover type between secondary forest and plantation or agriculture (Figure 2). Shrub may be less carefully managed than other land-covers and land ownership may be less established, meaning any fire is less likely to be quickly suppressed. In 2017, shrub covered 15% of Riau.

Natural forest areas that did not experience a land-cover transition experienced the lowest fire frequency compared to other land-covers, reflecting the low susceptibility of natural forests to fire. In particular, primary forests experienced a very low fire frequency, more than a factor of 60 less than shrub. Secondary forests experienced six times more fire than primary forests, but still a factor of 6–7 less than shrub. Protecting remaining primary forests is important, but these forests now cover less than 13% of Riau's forest. Secondary forest now accounts for 87% of natural forest in Riau. The moratorium on development of plantations on primary forests [26] will therefore only prohibit plantation development from a relatively small land area within Riau. Extending this moratorium to include secondary forests would likely lead to much larger reduction in fire. Our analysis further establishes the need to protect secondary forests, which have previously been shown to be important for carbon sequestration [27] and protection of biodiversity [28,29].

A link between land-cover change and fire has been shown previously. Analysis of Sumatran fires in 2013 found that 58% of fires occurred on land that had been forest 5 years previously [9]. Across Indonesia, 25% of forest loss in oil palm concessions experienced coincident fire the same year or

one year before forest loss [30]. In Riau, active fires were found to occur on average 58 ± 10 days before loss of natural forest [31], further confirming the very tight association between fire and forest loss. In Riau, fire frequency was a factor of 6 greater in regions that experienced forest canopy loss, compared to regions with no loss [16]. Analysis of 2015 fires in Sumatra found that rainfall, slope and population density were the most important variables in prediction of fires at regional and 1 km$^2$ pixel scale [32]. Fire management efforts in Indonesia need to consider the links between land-cover change and fire, the low fire frequency in undistubed natural forest and the higher fire frequency in degraded landscapes covered by shrub. Our analysis confirms that the Indonesian Peatland Restoration Agency plans to rewet and revegetate peatlands [33], should help to reduce the risk of fire.

Links between land-cover change and fire have also been demonstrated in the Amazon, with most fires in the 2000s linked to conversion of forest to agricultural land [34]. At a regional scale, there is a positive relationship between deforestation rate and fire emissions over the period 2001 to 2012 [35]. Over the period 1990 to 2014, Amazon-wide forest loss explained 31% of the variability in Amazon fire emissions [36]. Other studies have found that in the Rondônia and Mato Grosso regions of Amazonia, 53% of fires in 2005 occurred in land that had been deforested within the prior 5 years [37]. Areas of cleared Amazon broadleaf forest were very likely to burn shortly after forest cover loss, with 46% burning with 5 years [38]. An increased frequency of fire in regions with declining deforestation rate in primary forests, may be due to increased loss of secondary forest [39,40]. The extensive Amazon fires that occurred in 2019 have been linked to increased deforestation [41].

## 5. Conclusions

We combined information on land-cover transitions and the location and timing of fires to demonstrate the close connection between fire and land-cover change in Riau, Sumatra. Fires are a component of the land management process and are used to clear vegetation from the land. We found that areas that experienced a conversion in land-cover type, experienced more frequent fire than areas that did not change land-cover. In particular, we found the greatest fire frequency in areas of secondary forest that that were changed to shrub or converted to plantation. The peak in fire frequency occurred at the same time as the land-cover transition, confirming the close association between fire and land-cover change.

Frequent fire occurred in areas of shrub, which experienced fire frequency >60 times greater than primary forest and seven times greater than secondary forest. Plantation and agriculture experienced less fire than shrub, but still 17 times the rate in primary forest and double the rate of secondary forests. The conversion of natural forest to shrub, and to a lesser extent plantation and agriculture, has therefore created a fire prone landscape.

Efforts to reduce fire in Indonesia need to focus on the link between land-cover change and fire. Protecting remaining areas of natural forest, through establishing and maintaining adequately resourced protected areas, will help prevent further expansion of fire-prone shrub. Extending the plantation moratorium to include secondary forests as well as primary forests would also help reduce the conversion of natural forests and reduce the frequency of fire. Reducing the susceptibility of the landscape to fire, through restoring, rewetting and revegetating degraded shrub, particularly on peatlands, is a priority.

**Author Contributions:** Methodology—H.A.A., D.V.S., S.R.A. Formal Analysis—H.A.A. D.V.S., Visualization—H.A.A. Writing—original draft, H.A.A.; Writing—review and editing, D.V.S., S.R.A., I.S.S., L.S. All authors have read and agreed to the published version of the manuscript.

**Funding:** This research is part of a PhD study funded by the Indonesian Endowment Fund for Education (LPDP) and partly funded by the TransFoRM (Transboundary Fire Haze: Regional Characterization, Prediction and Mitigation in South-East Asia) project through the British Council Institutional Links (grant no. 332397925).

**Acknowledgments:** We are grateful to the Directorate of Forest Resources Inventory and Monitoring, Director General of Forestry Planning and Environmental Governance, Indonesian Ministry of Environment and Forestry for providing the land cover dataset.

**Conflicts of Interest:** The authors declare no conflict of interest.

## Appendix A

**Table A1.** Land Cover Classes [18].

| Class | Code | Description |
|---|---|---|
| Primary dryland forest | 2001 (Hp) | Natural tropical forests grow on non-wet habitat including lowland, upland, and montane forests with no signs of logging activities. The forest is including pygmies and heath forest and forest on ultramafic and lime-stone, as well as coniferous, deciduous and mist or cloud forest. |
| Secondary dryland forest | 2002 (Hs) | Natural tropical forest grow on non-wet habitat including lowland, upland, and montane forests that exhibit signs of logging activities indicated by patterns and spotting of logging. The forest is including pygmies and heath forest and forest on ultramafic and lime-stone, as well as coniferous, deciduous and mist or cloud forest. |
| Primary swamp forest | 2005 (Hrp) | Natural tropical forest grow on wet habitat including brackish swamp, sago and peat swamp, with no signs of logging activities. |
| Secondary swamp forest | 20051 (Hrs) | Natural tropical forest grow on wet habitat including brackish swamp, sago and peat swamp that exhibit signs of logging activities indicated by patterns and spotting of logging. |
| Primary mangrove forest | 2004 (Hmp) | Inundated forest with access to sea/brackish water and dominated by species of mangrove and Nipa (*Nipa frutescens*) that has no signs of logging activities. |
| Secondary mangrove forest | 20041 (Hms) | Inundated forest with access to sea/brackish water and dominated by species of mangrove and Nipa (*Nipa frutescens*) that exhibit signs of logging activities indicated by patterns and spotting of logging. |
| Plantation forest | 2006 (Ht) | Planted forest including areas of reforestation, industrial plantation forest and community plantation forest. |
| Non-Forest Dry shrub | 2007 (B) | Highly degraded log over areas on non-wet habitat that are ongoing process of succession but not yet reach stable forest ecosystem, having natural scattered trees or shrubs. |
| Wet shrub/swampy shrub | 20071 (Br) | Highly degraded log over areas on wet habitat that are ongoing process of succession but not yet reach stable forest ecosystem, having natural scattered trees or shrubs. |
| Savanna and Grasses | 3000 (S) | Areas with grasses and scattered natural trees and shrubs. This is typical of natural ecosystem and appearance on Sulawesi Tenggara, Nusa Tenggara Timur, and south part of Papua island. This type of cover could be on wet or non-wet habitat. |
| Dry Agriculture | 20091 (Pt) | All land covers associated to agriculture activities on dry/non-wet land, such as moor (*tegalan*), mixed garden and agriculture fields (*ladang*) |
| Mixed dry agriculture | 20092 (Pc) | All land covers associated to agriculture activities on dry/non-wet land that mixed with shrubs, thickets, and log over forest. Tis cover type often results of shifting cultivation and its rotation, including on karts |
| Estate crop | 2010 (Pk) | Estate areas that has been planted, mostly with perennials crops or other agriculture trees commodities |
| Paddy field | 20093 (Sw) | Agriculture areas on wet habitat, especially for paddy, that typically exhibit dyke patterns (*pola pematang*). Tis cover type includes rainfed, seasonal paddy field, and irrigated paddy fields |
| Transmigration areas | 20122 (Tr) | Kind of unique settlement areas that exhibit association of houses and agroforestry and/or garden at surrounding |

**Table A1.** *Cont.*

| Class | Code | Description |
| --- | --- | --- |
| Fish pond/aquaculture | 20094 (Tm) | Areas exhibit aquaculture activities including fish ponds, shrimp ponds or salt ponds |
| Bare ground/Bare soil | 2014 (T) | Bare grounds and areas with no vegetation cover yet, including open exposure areas, craters, sandbanks, sediments, and areas post fire that has not yet exhibit regrowth |
| Mining areas | 20141 (Tb) | Mining areas exhibit open mining activities such as open-pit mining including tailing ground |
| Settlement areas | 2012 (Pm) | Settlement areas including rural, urban, industrial and other settlements with typical appearance |
| Port and harbor | 20121 (Bdr/Plb) | Sighting of port and harbor that big enough to independently delineated as independent object |
| Open water | 5001 (A) | Sighting of open water including ocean, rivers, lakes, and ponds |
| Open swamp | 50011 (Rw) | Sighting of open swamp with few vegetation |
| Clouds and no-data | 2500 (Aw) | Sighting of clouds and clouds shadow with size more than 4 $cm^2$ at 100,000 scales display. |

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
