# Peer review of "Forest and Land Fires Are Mainly Associated with Deforestation in Riau Province, Indonesia"

_remotesensing, doi:10.3390/rs12010003_

Round 1

Reviewer 1 Report

The paper is well organized and well written. The problem is clearly described and interesting.

The authors describe the relations between land cover or land cover transition and "fire frequency". While the discussion of land cover is detailed and clear, I believe that the authors should clarify what they call "fire frequency". In captions of Figure 5 and Figure 6, they use instead "hot spot density", which must be the number of (high confidence) active fires per km2 and year. Is this what is called "fire frequency" across the paper? As pointed out by many authors, active fire density can be quite different from burned area, especially when observations are not possible due to clouds and smoke, which is a major factor over Indonesia and for peat fires in particular. Therefore, I think the authors should (1) clarify that "fire frequency" is "active fire density", and (2) discuss why "fire density" is an appropriate descriptor for the analysis of the relation between land cover and the incidence of fire.

In general the figures in the manuscript are well made and very clear. There is a detail, however, that makes them hard to read: the legend uses typically the 3-letter abbreviations described in Table 1, which are hard to understand. I suggest using longer and more self-explanatory abbreviations to help the reader.

I have a few specific comments:

L67. The authors indicate the footprint for active fires for vertical observations, but the footprint can be as large as 10km2 at the edge of scan. For a large footprint and, hence, high geolocation uncertainty, does the association between fire frequency and land cover still hold over the study area? This should be discussed.

L72: The sentence is difficult to understand. What does "categorize pixels" mean?

Figure1, Figure 2, Table 2, and Figure 3. The years that are displayed are not the same. Figure 1 displays all available years. Then, the authors focus on a subset of years to discuss the major land cover transitions. However, the years selected for Figure 2, Table 2, and Figure 3 do not match. Could the authors explain the criteria used for selecting years for each one of those representations?

L148. Authors group data by "land concessions" in Figure 6. It is not clear to me what is the source for this information since this is not discussed in Section 2.

L166. The authors conclude that there is a causal relation between fires and land-cover transition. I believe they should substantiate this claim more carefully. The analysis that was carried out shows that there is a clear relation but not that there is a causal relation.

Reviewer 2 Report

Line 38

The clearance clearing of forests [10-12] and drainage of peatlands, largely to establish oil palm and acacia plantations [13], has made the landscape more susceptible to fire.

Line 59    awkward wording -

During 2000-2009, analysis switched to using digital copies of Landsat images but the classification process still depended on a human interpreter.

For the 2000 – 2009 analyzes digital Landsat images were used, but the classification was still depending on visual image interpretation.

Line 60  It is unclear how adding the new sources of imagery would speed up the process.

In order to speed up the classification, 1000 m SPOT Vegetation and 250 m MODIS imageries images were used in addition to Landsat 7 ETM+.

Line 62 –  There is a mismatch of level of detail in the methods.  I am not sure the reader needs to know that the Landsat images are stored as a single geodatabase.  See other comments about methods. 

Finally, since 2009 only Landsat images have been used as main data source and stored as a single geodatabase (http://webgis.menlhk.go.id:8080/pl/pl.htm) and Landsat 8 OLI have been used since 2013.

Methods:  the methods describe only the source of the imagery for different time periods.  There is no description of the methods used to do the image interpretation, the assessment of the classification.  I would suggest rewriting the first paragraph to summarize the methods used in the national forest inventory - classification methods used in addition to the sources of imagery used.  There is a reference to the fact that the process changed from visual image interpretation to some other classification approach, but that is not described.  If there is a single document that describes that work it should be sighted. 

Author Response

This manuscript is a resubmission of an earlier submission. The following is a list of the peer review reports and author responses from that submission.